# Deep Dictionary Learning: Synergizing Reconstruction and Classification

**Shahin Mahdizadehaghdam, Ashkan Panahi, Hamid Krim & Liyi Dai**
{smahdiz,apanahi,ahk}@ncsu.edu & liyi.dai.civ@mail.mil

## Abstract

Deep dictionary learning seeks multiple dictionaries at different image scales to capture complementary coherent characteristics. We propose a method for learning a hierarchy of synthesis dictionaries with an image classification goal. The dictionaries and classification parameters are trained by a classification objective, and the sparse features are extracted by reducing a reconstruction loss in each layer. The reconstruction objectives in some sense regularize the classification problem and inject source signal information in the extracted features. The performance of the proposed hierarchical method increases by adding more layers, which consequently makes this model easier to tune and adapt. The validation of the proposed approach is based on its classification performance using four benchmark datasets and is compared to a CNN of similar size.

## 1 Introduction

Parsimonious data representation by learning overcomplete dictionaries has shown promising results in a variety of problems such as image denoising (Elad & Aharon (2006)), and image classification (Zhang et al. (2016)). In their proposed Sparse Representation-based Classification (SRC) (Wright et al. (2009b)) authors represent an image as a combination of a few images in the training dataset subsequently followed by a classifier based on these feature vectors. Proposed refinements led to task-driven dictionary learning (Mairal et al. (2012)) and Label Consistent K-SVD (LC-KSVD) (Jiang et al. (2013)) resulting in an overcomplete dictionary, for joint sparse representation, and classification.

The aforementioned dictionary learning methods are based on entire images for training, and can hence be computationally expensive also when training datasets are small, poor performance may ensue. Convolutional Neural Networks, on the other hand, learn the initial features from small image patches to build a hierarchal set of features at different scales. Contrary to conventional wisdom, several experimental studies (He et al. (2016), He & Sun (2015)) have reported difficulty in training deeper neural networks and increasing their depth eventually leads to decreased performance. This is partially due to the vanishing/exploding gradient effect during training. This problem persists despite mitigations such as batch normalization (He et al. (2016)).

To cope with CNN's fore-noted limitations, and to exploit the intrinsic deep structure of data, we propose a hierarchical approach to maximize the mutual information $I(\boldsymbol{X}^{*(s)}, \boldsymbol{Y})$ between the output of the last layer and the labels. The output of each layer, $\boldsymbol{X}^{*(r)}$ $r \in \{1, .., s\}$, is obtained by maximizing its mutual information $I(\boldsymbol{X}^{*(r)}, \boldsymbol{X}^{*(r-1)})$ with the associated input of the layer. We have shown that under a certain generative model, this information driven approach is equivalent to a hierarchical (deep) dictionary learning approach which achieves optimal representations of input signals for classification, while ensuring a maximal mutual information with the original signal.

Within this framework, the front layer dictionary is learned on small image patches, and the subsequent layer dictionaries are learned on larger scales. Put simply, the initial scale captures the fine low-level structures comprising the image vectors, while the next scales coherently capture more complex structures. The classification is ultimately carried out by assembling the final and largest scale features of an image and assessing their contribution. In contrast to CNN, we show that the performance of the proposed DDL method improves with additional layers, hence indicating an

amenability to tuning, and a better potential for more elaborate learning tasks such as transfer learning.

## 2 PROPOSED APPROACH

As explained in Section 1, in addition to seeking the maximum mutual information with the labels, our method additionally seeks to attain maximum mutual information between the feature vectors and the original signals. To this end, and from an information theoretic point of view, we characterize our classifier as problem ($P1$),

$$\theta^* = \arg\max_{\theta} \ I(\boldsymbol{X}^{*(s)}, \boldsymbol{Y}) \qquad\qquad (P1)$$

$$st: \boldsymbol{X}^{*(r)} = \arg\max_{\boldsymbol{X}^{(r)}} \ I(\boldsymbol{X}^{(r)}, \boldsymbol{X}^{*(r-1)}), \ r \in \{1, .., s\}, \qquad (1)$$

where $I(\boldsymbol{X}^{*(s)}, \boldsymbol{Y})$ is the mutual information between the optimal feature vectors from the last layer, $\boldsymbol{X}^{*(s)}$, and the labels, $\boldsymbol{Y}$, and $\theta$ is the classification model parameter. Problem (P1) seeks to maximize over $\theta$, the mutual information between the labels and the features of layer $\boldsymbol{X}^{*(s)}$. By virtue of the constraint, this results in a maximal mutual information between input and output of a layer and hence also sequentially secure this maximal information property with the input signal $\boldsymbol{X}^{*(0)}$. By assuming a certain generative model we can show that ($P1$) is equivalent to the following problem,

$$\{\boldsymbol{W}^*, \{\boldsymbol{D}^{*(r)}\}_{r=1}^{s}\} = \arg\min_{\boldsymbol{W}, \ \{\boldsymbol{D}^{(r)}\}_{r=1}^{s}} \ \mathcal{L}^{\mathcal{C}}(\boldsymbol{Y}, \boldsymbol{X}^{*(s)}, \boldsymbol{W}), \qquad (2)$$

where $\mathcal{L}^{\mathcal{C}}$ is a desired classification loss functional, and $\boldsymbol{W}$, $\boldsymbol{D}^{(r)}$, and $\boldsymbol{Y}$ are respectively the classification parameters, the dictionary for layer $r$, and the labels of the training images. $\boldsymbol{X}^{*(s)}$ is the input to the classifier. These vectors are the result of the following recursive relation:

$$\boldsymbol{X}^{*(r)} = \arg\min_{\boldsymbol{X}^{(r)}} \ \mathcal{L}_r^{\mathcal{R}}(\boldsymbol{D}^{(r)}, \boldsymbol{X}^{(r)}, \boldsymbol{X}^{*(r-1)}), \ r \in \{1, .., s\},$$

$$s.t.: \ \mathcal{L}_r^{\mathcal{R}}(\boldsymbol{D}^{(r)}, \boldsymbol{X}^{(r)}, \boldsymbol{X}^{*(r-1)}) = \frac{1}{2}||\boldsymbol{X}^{*(r-1)} - \boldsymbol{D}^{(r)}\boldsymbol{A}^{(r)}||_F^2 + \lambda||\boldsymbol{A}^{(r)}||_1, +\lambda'||\boldsymbol{A}^{(r)}||_F^2, \quad (3)$$

$$\boldsymbol{X}^{(r)} = P_r(\boldsymbol{A}^{(r)}), \ \ \boldsymbol{D}^{(r)} \in \mathcal{C}, \ \ \boldsymbol{A}^{(r)} \geq 0,$$

where each column of the matrix $\boldsymbol{X}^{*(0)}$ is a vectorized image patch, and $P_r$ is an operator which concatenates the feature vectors of adjacent patches. In other words, $P_r$ is an operator which reshapes $\boldsymbol{A}^{(r)}$ as the output of layer $r$ (input to the layer $(r+1)$). Eqn. (3) is similar to an elastic net regularization problem with a non-negativity condition on the feature vectors. We emphasize that the objective functional in Eqn. (2) implicitly depends on all dictionaries, since the computation of $\boldsymbol{X}^{*(s)}$ in Eqn. (3) invokes all of them.

We show in Fig. 1 the sequence of computational steps exploiting matrices from Eqn. (3) and their associated structures. The white arrows in the figure depict the forward computing path of the sparse representation of each input layer. A set of images are first segmented into small image patches and vectorized into matrix $\boldsymbol{X}^{*(0)}$, the input of the first layer. For a given first layer dictionary $\boldsymbol{D}^{(1)}$, the sparse representations of the image patches are learned as the columns of the $\boldsymbol{A}^{*(1)}$ matrix, by solving Eqn. (3) via the sparse encoding algorithm FISTA (Beck & Teboulle (2009)). The patches of $\boldsymbol{A}^{*(1)}$ ($3 \times 3$ window) are next reshaped by operator $P_1$ to yield input $\boldsymbol{X}^{*(1)}$ to the second layer. This analysis is sequentially carried on to scale $s$ where $\boldsymbol{A}^{*(s)}$ yields $\boldsymbol{X}^{*(s)}$ as an input to a classifier (such that each column of this matrix represents an image). The class label information of the images in matrix $\boldsymbol{Y}$, together with $\boldsymbol{X}^{*(s)}$ provide the classification parameters $\boldsymbol{W}$ as a solution to Eqn. (2).

The blue arrows in Fig. 1 highlight the backward training path, and reflect the updates on the dictionaries (by way of gradient descent on Eqn. (2)), as a result of the optimized classification. Optimizing the classification loss $\mathcal{L}^C$ in the forward pass (path of Fig. 1), is followed by updates on the backward pass (path of Fig. 1).

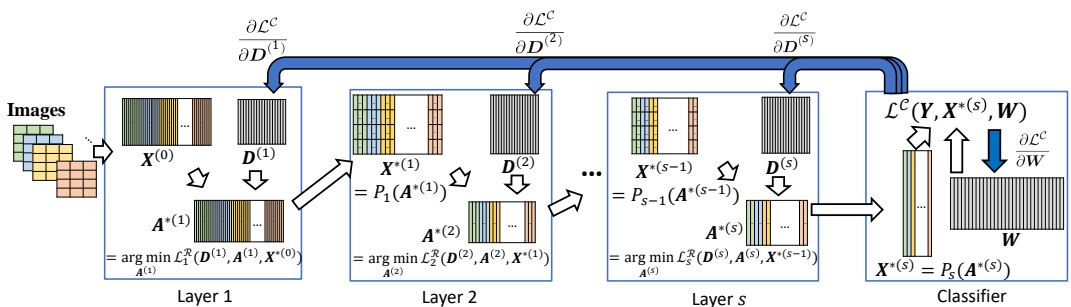

Figure 1: Sequential steps of a deep dictionary with $s$ layers.

In summary, the relevant parameter vectors of the images are learned through multiple forward-backward passes through the layers. The dictionaries are updated so that the resulting representations are suitable for the classification. Moreover, the representations are learned/regularized by minimizing the reconstruction loss at each layer.

## 3 EXPERIMENTS

In Table 1, we compared the performance of our proposed algorithm with the state of the art methods on a handwritten digit dataset, MNIST LeCun et al. (1998), and a face dataset, Extended YaleB Georghiades et al. (2001). Table 2, compares the accuracy of the proposed method with other state of the art methods which all have approximately a comparable number of parameters on CIFAR-10 and CIFAR-100. We also trained 4 different networks with a different number of layers to test their performance on the CIFAR-10 dataset. As Fig. 2 shows, the accuracy of our proposed approach increases as the structure depth increases.

Table 1: Classification accuracy on MNIST and Extended YaleB

| Method | MNIST | Extended YaleB |
|---|---|---|
| SRC Wright et al. (2009a) | 95.69 | 97.2 |
| LLC Wang et al. (2010) | 94.32 | 90.7 |
| LC-KSVD Jiang et al. (2013) | 92.58 | 96.7 |

Table 2: Classification accuracy on CIFAR-10 and CIFAR-100

| Method | Params | CIFAR-10 | CIFAR-100 |
|---|---|---|---|
| All-CNN Spring. et al. (2014) | ≈1.4M | 92.75 | 66.29 |
| CKN Mairal et al. (2014) | ≈0.32M | 78.30 | - |
| ResNet He et al. (2016) | ≈0.85M | 93.59 | 72.78 |
| **DDL 9-layers** | ≈1.4M | **93.04** | **68.76** |
| **DDL 15-layers** | ≈0.76M | **94.17** | **80.62** |

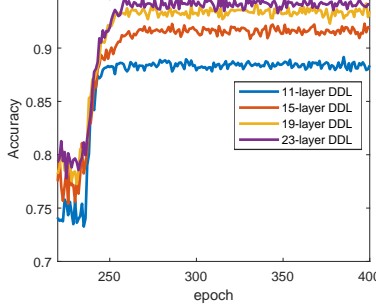

Figure 2: Learning curves of DDL with different number of layers on the CIFAR-10 dataset.

## 4 CONCLUSIONS

In this paper, we demonstrated the importance of representing images by learning image characteristics at multiple scales via deep dictionaries. The importance of preserving the maximum mutual information between the feature vectors and the input signals is rationalized from an information theoretic principle, and is tested empirically. We also showed that by increasing the depth of representation, the extracted feature vectors are maintaining high mutual information with the input data, thereby facilitating the training of the deep dictionaries in contrast to the Convolutional Neural Networks.

### ACKNOWLEDGMENTS

We would like to acknowledge the support of U.S. Army Research Office: Grant W911NF-16-2-0005.

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
