# OpenReview forum: "DEEP DICTIONARY LEARNING: SYNERGIZING RECONSTRUCTION AND CLASSIFICATION"
_ICLR.cc/2018/Workshop — Reject_

### Official Review · AnonReviewer2 · 2018-03-08
**Interesting direction but not clear how novel the architecture is. Not clear why it works.**

**Rating:** 5
**Confidence:** 3

**Review:**

I would like to encourage the authors work in deep dictionary learning. This is an area that has been studied quite some time and still probably do not have a clear thorough breakthrough comparable to CNNs.

I liked the authors' method minimizing the mutual information. Is this the part helping vanishing gradients for CNNs? The proposed architecture does not seem to be very novel. This architecture has been discussed/experimented maybe for about 7-8 years now. What makes the proposed method having decent results is not clear to me. Results seem interesting but not clear where the win comes from.

I did not understand Table 1, where the proposed method in that table?

I would have loved to see on the same data set where CNN gets broken with increased number of layers.

Minor: Paper structure was a little interesting. Introduction explained the proposed method (last two paragraphs). Then proposed method section came.

---

### Official Review · AnonReviewer4 · 2018-03-17
**Needs to be clearer where results coming from**

**Rating:** 3
**Confidence:** 1

**Review:**

This paper reports significantly better than state-of-the-art results on CIFAR-100 using a dictionary learning based method. However, I'm not seeing where the paper is adding a significantly new insight into using dictionary based methods, which aren't a new idea. There isn't enough detail to be sure that proper experimental practice was followed. Including code isn't a requirement for ICLR -- however, given past history on this sort of claim, I'd need more evidence and insight into what's new to believe these results. If the results do hold up, there should be no problem publishing an extended version of this paper at a top conference some time soon.

---

### Decision · Program_Chairs · 2018-03-20
**ICLR 2018 Workshop Acceptance Decision**

**Decision:**

Reject

**Comment:**

Based on the reviews, this paper has not been accepted for presentation at the ICLR workshop. However, the conversation and updates can continue to appear here on OpenReview.